# An Emerging Role for Anti-DNA Antibodies in Systemic Lupus Erythematosus

**DOI:** 10.3390/ijms242216499

**Published:** 2023-11-19

**Authors:** Tetsuo Kubota

**Affiliations:** Department of Medical Technology, Tsukuba International University, Tsuchiura 300-0051, Ibaraki, Japan; kbtmtec@tmd.ac.jp

**Keywords:** anti-DNA antibodies, systemic lupus erythematosus (SLE), penetration, endocytosis, neutrophil extracellular traps (NETs), NETosis, pathogenesis

## Abstract

Anti-DNA antibodies are hallmark autoantibodies produced in systemic lupus erythematosus (SLE), but their pathogenetic role is not fully understood. Accumulating evidence suggests that some anti-DNA antibodies enter different types of live cells and affect the pathophysiology of SLE by stimulating or impairing these cells. Circulating neutrophils in SLE are activated by a type I interferon or other stimuli and are primed to release neutrophil extracellular traps (NETs) on additional stimulation. Anti-DNA antibodies are also involved in this process and may induce NET release. Thereafter, they bind and protect extracellular DNA in the NETs from digestion by nucleases, resulting in increased NET immunogenicity. This review discusses the pathogenetic role of anti-DNA antibodies in SLE, mainly focusing on recent progress in the two research fields concerning antibody penetration into live cells and NETosis.

## 1. Introduction

Systemic lupus erythematosus (SLE) is a prototypic systemic autoimmune disease that preferentially affects women 20–40 years of age. Although clinical manifestations are varied, ranging from mild to severe, SLE often begins with a fever, skin rash, or arthritis and develops organ lesions such as serositis and glomerulonephritis or produces neuropsychiatric symptoms (NPSLE) [1,2]. Were it not for appropriate diagnosis and treatment, the organ lesions could leave patients severely disabled. Multiple genetic susceptibility and environmental factors are thought to lead to a breakdown of immunological self-tolerance, and different autoantibodies against nuclear antigens are detected in the serum [3]. Many SLE-susceptibility genes have been linked to type I interferon (IFN) production or responses, and therefore, numerous studies have been carried out to understand the “IFN signature” in SLE. So far, type I IFNs have been implicated in a loss of tolerance, the activation of neutrophils and the release of neutrophil extracellular traps (NETs), the production of the B-cell activating factor (BAFF), and other events; nevertheless, our understanding of the pathophysiology of SLE is still incomplete [4].

Among the antinuclear antibodies (ANA), those reactive with the double-stranded (ds)DNA and Sm nucleoprotein are relatively specific for SLE and are included in the classification criteria for this disease proposed by the European League Against Rheumatism (EULAR) and the American College of Rheumatology (ACR) [5]. According to current criteria, the detection of ANA at a titer of 1:80 or higher on HEp-2 cells is adopted as an entry criterion, and the presence of the anti-dsDNA antibody or anti-Sm antibody is weighed heavily in the additive immunology domain criteria. In typical cases, serum titers of anti-DNA antibodies correlate with disease activity, and they are regularly monitored over clinical follow-up. However, despite many efforts, our understanding of the pathogenetic role of these anti-DNA antibodies in SLE remains incomplete. This review discusses how anti-DNA antibodies are involved in lupus pathogenesis, mainly focusing on the following two issues: antibody penetration into live cells and relevance to NETosis are both issues that have been intensively studied recently.

## 2. Generation of Anti-DNA Antibodies

The analysis of frozen serum samples stored in a huge repository has revealed that, in many patients, anti-DNA antibodies are present a few years before the diagnosis of SLE [6]. Because the production of IgG anti-dsDNA antibodies is T-cell dependent, the activation of both autoreactive B cells and autoreactive T cells is necessary for this process [7]. However, native dsDNA itself is not immunogenic, and how patients with SLE consistently produce anti-DNA antibodies remains an open question. In one study, DNA was exogenously added to the cultures of HEK 293T cells that had been transfected with the gene for the SLE susceptibility allele HLA-DR15, which was internalized and then expressed on the cell surface together with this MHC class II molecule [8]. These investigators created NFAT-GFP reporter cells that were transfected with anti-DNA B cell receptors and expressed GFP and IL-2 upon the crosslinking of the receptors. When cocultured with the above-mentioned DNA presenting cells, the reporter cells were activated to produce GFP and IL-2. MHC class II molecules generally present peptide antigens to helper T cells, but this study proposes the unexpected role of MHC molecules in the activation of DNA-reactive B cells.

The generation of monoclonal antibody-producing hybridomas using human peripheral blood lymphocytes is difficult and usually yields solely low-affinity IgM antibodies. However, recent advances in molecular technology have facilitated the production of human monoclonal anti-DNA antibody-like proteins via the transfection of HEK 293T cells with immunoglobulin heavy chain genes identified from a single B cell from the peripheral blood of a patient with SLE [9]. Applying this technique to analyze the variable region gene’s use of anti-DNase1L3 neutralizing antibodies, interestingly, some were found to have been derived from anti-DNase1L3 germline-encoded precursors which acquired cross-reactivity to dsDNA following somatic hypermutation [10]. Another study reported that some mouse anti-dsDNA monoclonal antibodies were cross-reactive with spermatid nuclear transition protein 1 [11]. These studies suggest the possibility that anti-DNA antibodies might initially be produced in response to unexpected DNA-binding protein antigens.

## 3. Penetration of Anti-DNA Antibodies into Live Cells (Figure 1)

The ability of ANA to enter the nucleus of live cells was initially reported by Alarcón-Segovia et al. in 1978 [12]. Using a direct immunostaining method without a second antibody, they documented the internalization of anti-RNP antibodies obtained from a patient with mixed connective tissue disease into normal peripheral blood mononuclear cells (PBMCs). Soon after, they reported similar findings with anti-DNA antibodies as well [13]. Initially, these findings met with skepticism, but gradually, many studies confirmed this phenomenon [14,15,16]. The mechanisms responsible for internalization are multifarious. Some anti-DNA antibodies enter cells via Fc-receptor-mediated endocytosis, but there are examples showing that recombinant single-chain fragments of the variable chains (scFv) lacking the Fc region can still enter cells [17,18]. Some anti-DNA antibodies enter the nucleus and bind to chromatin DNA, while others remain in the cytoplasm: the factors that determine how much movement remains unidentified.

**Figure 1 ijms-24-16499-f001:**
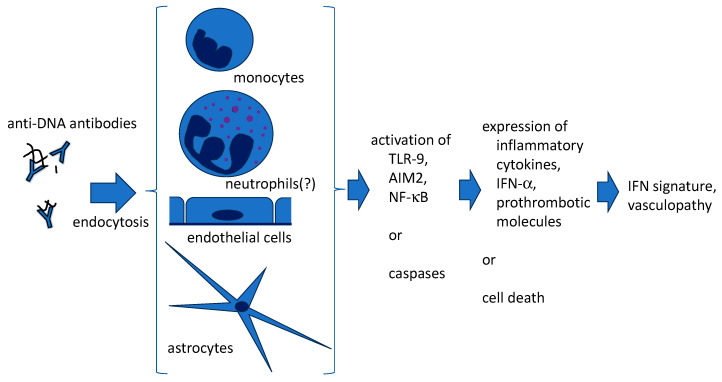
The internalization of anti-DNA antibodies via living cells may affect the pathophysiology of SLE. Apart from immortalized or genetically modified cell lines, anti-DNA antibodies are also demonstrated to enter several normal cell types, accompanied by DNA. As a result, cells are activated to produce lupus-prone cytokines, prothrombotic molecules, or might be impaired.

Anti-DNA antibodies form immune complexes with DNA in vivo in the plasma or in vitro in a culture medium. Although these immune complexes would be trimmed using DNase, when researchers use pure antibodies, they must be washed thoroughly in a high salt buffer and/or alkaline buffer [19,20]. Because the ratio of absorbance at 260 nm and 280 nm changes only slightly but significantly before and after washing, it is speculated that, for example, without sufficiently thorough washing, short oligonucleotides could remain attached to the antigen-binding cleft of the antibodies purified using the protein G column. Even after the preparation of ultra-purified antibodies, however, they still bind again to DNA in the medium or on the cell surface when added to cell cultures. Therefore, even very highly purified anti-DNA antibodies should be considered immune complexes in most studies, even without the addition of exogenous DNA.

In parallel with the discovery of various intracellular nucleic acid sensors, it has been suggested that the DNA that enters the cells accompanying anti-DNA antibodies stimulates Toll-like receptors (TLRs) or other nucleic acid sensors expressed in the endosome or in the cytosol, leading to the production of cytokines relevant to lupus pathogenesis [21,22]. In line with this, our laboratory shows that the mouse monoclonal antibody 2C10, which specifically recognizes dsDNA and does not bind to single-stranded (ss)DNA, enters the nucleus of PBMCs from healthy subjects and induces the expression of cytokines commonly implicated in lupus, including IFN-α, IFN-β, TNF-α, IL-1β, and MCP-1 [23]. The internalization of 2C10 is significantly inhibited by the macropinocytosis inhibitor cytochalasin D but not by an Fcγ-receptor blocker. Cytokine expression was suppressed by cytochalasin D and the TLR-9 inhibitor chloroquine. In addition, the NLRP3 inhibitor shikonin suppressed the secretion of certain cytokines, including IL-1β. These results suggest that 2C10 was endocytosed mainly by monocytes via macropinocytosis, and the accompanying DNA ligated TLR-9 in the endosome, and after leaking into the cytosol, stimulated AIM-2. Another monoclonal anti-DNA antibody, WB-6, which is cross-reactive with dsDNA, ssDNA, and cardiolipin-β_2_GPI, was observed to enter normal monocytes and induce tissue factor expression [20,24]. Since internalization was diminished by the pretreatment of cells with DNase 1, WB-6 was suggested to enter cells by binding to the cell surface’s DNA. As a result, WB-6 stimulated not the TLR-4 axis, which has been suggested to be the major route in previous studies [25], but the TLR-9 pathway, leading to tissue factor expression and a prothrombotic state in a mouse model [26].

Clinical phenotypes of NPSLE are diverse and are classified into neurological syndromes (including headache, seizure disorders, and cerebrovascular disease) and diffuse psychiatric or neuropsychological syndromes (including cognitive impairment, mood disorder, anxiety disorder, and psychosis) [27]. At least some neurological symptoms are ascribed to the pathological effects of antiphospholipid antibodies (aPL) on the vascular system. Although it is hypothesized that some autoantibodies are involved, the pathogenetic role of autoantibodies in diffuse psychiatric or neuropsychological syndromes remains undefined [28]. In addition to the blood–brain barrier, however, several other interfaces may serve as sites of antibody transfer into the central nervous system (CNS), such as the meningeal barrier, the glymphatic pathway, and the blood–cerebrospinal fluid barrier; the permeability of these barriers is considered to increase under pathological conditions [27]. It is noteworthy that Stamou et al. [29] documented the internalization of IgG-anti–IgG immune complexes by newborn rat hippocampal cells via Fcγ receptors. Based on these findings, we tested whether 2C10 enters cells of the CNS and found that it does enter the nucleus of rat astrocytes, but not neurons, in in vitro cultures [30]. The effects of 2C10 internalization on the function of astrocytes have not yet been determined, but given the pivotal role of astrocytes in regulating brain activity [31], they might be relevant to the pathogenesis of diffuse psychiatric or neuropsychological syndromes in NPSLE.

Using the well-studied mouse monoclonal anti-DNA antibodies 3D8 and 3E10, molecular mechanisms of cell penetration have been explored and reviewed in detail [32]. Briefly, following the binding to the cell surface, the heparan sulfate proteoglycan, 3D8, is engulfed into early endosomes and then dissociates from the heparan sulfate, changes its conformation, and escapes into the cytosol. By contrast, 3E10 is proposed to enter the cells via a mechanism not involving endocytosis but in a manner dependent on equilibrative nucleoside transporter 2 (ENT2). ENT2 is an integral membrane protein widely expressed in most cell types, playing a role in transporting nucleosides. The knockdown of ENT2 or adding the ENT2 inhibitor dipyridamole reduces the penetration of 3E10 into the cell. Further, 3E10 traffics to the nucleus via an uncertain mechanism. It is noteworthy that in a mouse model, a dimeric scFv structural 3E10 variant (designated DX1) was suggested to be transcytosed through the endothelial cells of the brain and, thus, cross the blood–brain barrier [33]. Dipyridamole reduced the transfer of DX1. Such a study aimed to develop antibody-based immunotherapy for brain tumors could also be relevant to the pathological mechanism of NPSLE. Antibody transcytosis across the brain’s endothelial cells is a hot topic [34], and it would be intriguing to explore the molecular mechanisms of how DX1 interacts with ENT2 and enters and exits the brain’s endothelial cells.

## 4. Anti-DNA Antibodies and NETs

### 4.1. What Are NETs?

Eight years after Takei et al. [35] described characteristic morphological changes in neutrophils stimulated by phorbol 12-myristate 13-acetate, Brinkmann et al. [36] described the basic structure and antimicrobial function of NETs using impressive electron microscopy. Since these publications, NET release has been recognized as a new type of cell death by neutrophils mediating antimicrobial suicide attacks [37]; this process has been designated NETosis. NETs are web-like structures released by neutrophils and triggered by different stimuli; they are composed of DNA originating from decondensed chromatin or the mitochondria and are decorated by histones, high mobility group box 1 (HMGB1), and various neutrophil granular antimicrobial proteins or peptides including myeloperoxidase (MPO) and LL37 (37 amino acid residues of the C-terminal region of a human cationic antimicrobial protein, hCAP). Although a similar extracellular trap (ET) formation has also been observed in mast cells [38] and eosinophils [39] playing a role in innate self-defense mechanisms, NETs have been most intensively studied recently in the context of autoimmunity.

### 4.2. NETs in Autoimmune Diseasses

As well as in infectious diseases, including most recently in COVID-19 [40], NETs are known to be triggered by a variety of sterile stimuli and are involved in autoimmune diseases, including SLE, antiphospholipid syndrome (APS), anti-neutrophil cytoplasmic antibody (ANCA)-associated vasculitis (AAV), and rheumatoid arthritis [41].

#### 4.2.1. SLE

Peripheral blood neutrophils obtained from SLE patients release more NETs than those from healthy donors triggered by different stimuli or even spontaneously in ex vivo experiments. Lande and colleagues [42] showed that NETs in the sera obtained from SLE contain DNA, anti-DNA antibodies, and the antimicrobial peptides LL37 and HNPs (human neutrophil peptides belonging to the α-defensin family). These complexes of DNA, anti-DNA, and peptides stimulated normal plasmacytoid dendritic cells (pDCs) to produce IFN-α through the TLR-9 pathway. Interestingly, DNA-anti-DNA immune complexes alone did not stimulate pDCs, and LL37 and/or HNPs were necessary to activate them. It was suggested that LL37 induces the aggregation of DNA fragments to form insoluble particles that are resistant to nuclease digestion and enable the DNA to enter the intracellular TLR-9-containing compartments of pDCs. In the same issue of that journal, another study focusing on pediatric SLE found that healthy neutrophils showed increased levels of TLR-7 mRNA after exposure to SLE sera or IFN-α [43]. Accordingly, SLE neutrophils produced significantly high levels of IL-8 in response to a TLR-7 agonist. These observations prompted the researchers to assess the effect of anti-RNP antibodies. They found that SLE neutrophils, presumably primed in vivo by IFN-α, showed NETosis after 3 h of culture with IgG anti-RNP antibodies purified from SLE serum in a FcγRIIa-, NADPH-, and TLR-7-dependent manner.

To prevent NET release in SLE, a study tested the effect of ligation on one of the negative regulators of the neutrophil function with the signal inhibitory receptor on leukocytes-1 (SIRL-1) [44]. Via ligation with anti-SIRL-1 antibodies, spontaneous and anti-LL37 antibody-induced NET release by SLE neutrophils was significantly suppressed. These results suggest that NET release could be a strategically important therapeutic target in SLE.

#### 4.2.2. APS

NETs participate in a prothrombotic state via multiple mechanisms, including the inhibition of tissue factor pathway inhibitors, the activation of platelets, the activation of procoagulant factors, and the induction of activated protein C (APC) resistance [45,46]. The frequency of aPL-positive patients in SLE is estimated to be 30–40% [47]. Not all of these patients exhibit aPL-related clinical manifestations, but they have a higher risk of vascular events than aPL-negative patients. By contrast, about half of the patients with APS have secondary APS, which is mostly associated with SLE. Thus, there is a significant overlap in the pathological condition of SLE and APS.

IgG purified from patients with primary APS and human IgG monoclonal anti-β_2_GPI antibodies both induced NET release from normal neutrophils in one study [48]. These investigators reported that anti-β_2_-GPI antibodies likely bind to the cell surface β_2_-GPI and thereby stimulate the cells. Furthermore, in vivo testing of APS IgG in a mouse model resulted in exaggerated thrombosis with thrombi enriched for citrullinated histone H3 (a marker of NETs) [49]. Large amounts of human IgG were bound to the surface of the mouse neutrophils. Although endothelial cells, platelets, and monocytes are the main players involved in APS pathogenesis, these reports reveal the important role of neutrophils as well. However, it remains uncertain how and to what extent β_2_-GPI is expressed on the normal neutrophil surface. In one study on large cohorts of patients with SLE, secondary APS associated with SLE, or primary APS, the NET-release triggering activity of patient’s plasma samples on healthy neutrophils was compared [50]. The results showed that plasma samples collected from 60% of SLE, 61% of SLE + APS, and 45% of primary APS patients were able to induce NET release.

#### 4.2.3. AAV

Different forms of NETosis are observed depending on how they are triggered. Two major types, late suicidal NETosis (also referred to as lytic NET formation) and early vital NETosis (rapid non-lytic NET formation), are reviewed elsewhere [51]. Late suicidal NETosis depends on the production of reactive oxygen species by NADPH-oxidase and takes a few hours. Unfolded chromatin is released into the cytosol, decorated with granular and cytosolic proteins, and is finally expelled by plasma membrane disruption. In contrast, vital NETosis occurs within minutes of the stimulation, independently of oxidants, and NETs are released by nuclear–envelope blebbing and a vesicular export without the rupture of the plasma membrane; thus, the cells remain alive.

The characteristics of NETosis induced in healthy neutrophils by patient sera were compared in large cohorts of ANCA-positive AAV (*n* = 80) and ANA-positive SLE (*n* = 59). Interestingly, the incubation of healthy neutrophils with AAV sera induced late suicidal NETosis, whereas SLE sera induced vital NETosis [52]. AAV-induced NETosis was triggered independently of IgG, whereas SLE-induced NETosis was dependent on Fcγ receptor signaling. Soluble IgG isolated from SLE sera did not induce NETosis, but immobilized SLE-IgG, which mimics immune complexes, did. These results suggest that vital NETosis requires the intensive cross-linking of Fcγ receptors.

### 4.3. Quantification of NETs

The quantification of NETs is challenging due to their varied morphology, heterogeneous components, and especially their fragility. A possible simple approach to detect NETs in plasma or in the culture medium is by using a sandwich ELISA with a capture antibody such as anti-MPO and a detection antibody such as anti-DNA. This may be satisfactory for certain assays, with accepted limitations, but developing a reliable, generally applicable ELISA system is not straightforward [53]. Recently, an improved highly sensitive ELISA protocol was proposed, using two different antibodies (anti-MPO and anti-citrullinated histone H3) for capture and one (anti-DNA) for detection [54]. In parallel, a simple, inexpensive, immunofluorescence smear assay was developed, in which 1 µL of SLE plasma was fixed onto poly-L-lysine-coated glass slides, followed by staining extracellular DNA with SYTOX Green and DAPI. Fluorescence intensity was quantified using image analysis software (ImageJ, https://imagej.net/ij/), and the assay results were shown to correlate well with the ELISA. An assay using a more sophisticated imaging system employing three-dimensional immunofluorescence confocal microscopy is published as a video article; this technique can be utilized to observe the process of the formation and degradation of NETs quantitatively [55].

When we consider introducing an assessment of NETs into the clinical laboratory, it is more practical to measure some NET-associated serological markers in place of the challenge of detecting the whole structure of NETs. One study has indicated that SLE patients have higher cell-free DNA, MPO activity, anti-MPO antibodies, DNase I concentration, and lower NETolytic activity compared to the healthy controls [56]. These changes in NET-related parameters were shown to be correlated with disease activity.

### 4.4. Anti-NET Antibodies

It is plausible that NETs are antigenic because molecules normally contained in the nucleus or granules are extruded, may be in a modified form, and are exposed to the immune system for a long period due to decreased degradation activity. In fact, the production of various autoantibodies reactive to NET components has been reported.

In one study, anti-NET antibodies were detected using indirect immunofluorescence in 10 of 19 patients with microscopic polyangiitis [57]; these antibodies were distinct from ANCA, but their target antigens were not determined. It is noteworthy that ANA was negative in all these anti-NET-positive patients, indicating that fundamental self-tolerance mechanisms were not entirely absent. In another study, IgG and IgM antibodies to NETs measured by ELISA were significantly elevated in patients with primary APS and in SLE without aPL relative to the healthy controls [58]. In a recent larger cohort of primary APS, 45% of the aPL-positive patients had IgG and/or IgM anti-NET antibodies [59]. Importantly, an analysis of the associations of anti-NET antibodies with clinical manifestations revealed that IgG antibodies were associated with lesions affecting the white matter of the brain, while IgM antibodies were tracked with complement consumption. Antigen specificities of these antibodies were analyzed using a 120-antigen microarray panel, and it was suggested that IgG antibodies to NETs were likely to be driven via their reactivity with protein antigens in the NETs, while the IgM antibodies to NETs were likely to target DNA. In another study, anti-NET antibodies were detected in 35.7% of patients with SLE [60]. Interestingly, 37.0% of these patients were negative for anti-dsDNA antibodies, indicating that DNA is not necessarily a major antigen in NETs, even in SLE.

Similar to antibodies that bind to DNA- or RNA-binding proteins such as histone, Sm, or RNP, SLE patients possess antibodies to antimicrobial DNA-binding peptides LL37 and HNPs in NETs [42]. In addition, the DNA sensors AIM2 and IFI16, which are released from dying cells and are present in plasma, bind to extracellular DNA in NETs; antibodies against AIM2 and IFI16 are also produced in SLE [61]. Furthermore, extracellular DNA–MPO-AIM2/IFI16 complexes were detected in the biopsy specimens of diffuse proliferative lupus nephritis, suggesting a distinct immunostimulatory role of AIM2 and IFI16 in the renal lesions.

Even though titers of anti-DNA antibodies correlate with such antibodies to NET components, there have been no reports which directly demonstrate that NETs induce the production of anti-DNA antibodies. It is possible that oxidized DNA present in NETs [62], which is known to be immunogenic [63], acts as a primary antigen, triggering the production of antibodies cross-reactive to native DNA. However, it is generally recognized that anti-DNA antibodies have been produced in SLE sometime before the occurrence of pathological conditions with NET release. Thus, not all NET components induce autoantibody production. For example, SLE patients do not produce anti-MPO antibodies. Conversely, AAV patients do not produce anti-DNA antibodies. What controls the antigenicity of the NET components has not been clearly explained.

### 4.5. Amplification of SLE Disease Activity by Anti-DNA Antibodies and NETs (Figure 2)

As discussed above, NET formation is increased in SLE. In the following, representative findings that are informative for the mechanisms responsible for the aggravation of NET and the pathological condition in SLE are reviewed.

**Figure 2 ijms-24-16499-f002:**
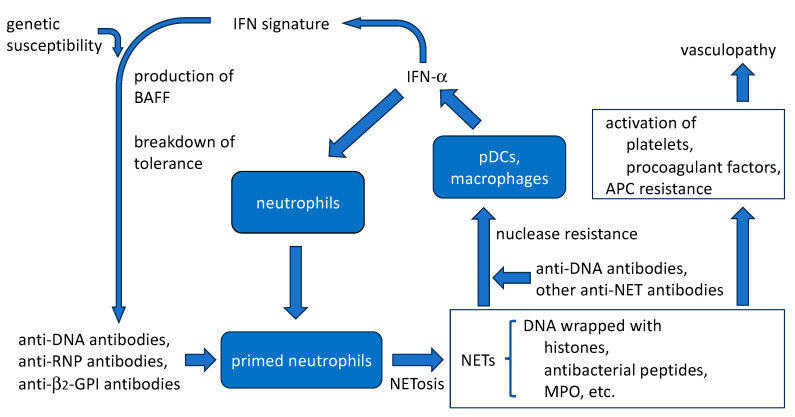
Anti-DNA antibodies are involved in the process of NETosis. They likely induce SLE neutrophils which have been primed by IFN-α, to release NETs. They also bind to extracellular DNA in NETs, which could enhance the immunogenic activity of the NETs.

#### 4.5.1. Aggravation of IFN Signature

Culturing pDCs isolated from healthy donors with apoptotic or necrotic neutrophils does not result in their activation. By contrast, culturing with NETs induces the production of IFNα by pDCs in a DNA- and TLR-9-dependent manner [42,43]. In another study, NETs were phagocytosed by macrophages and translocated from the phagosome to the cytosol, where they activated cGAS, leading to the production of type I interferon [64]. Furthermore, immune complexes of a panel of human monoclonal anti-DNA antibodies and NETs were suggested to be internalized by monocytes and endothelial cells in an Fc receptor-dependent manner, resulting in the enhanced expression of type I IFN and NF-κB, respectively [65].

#### 4.5.2. Protection of NETs from Nucleases by Anti-DNA Antibodies

DNA in the NETs is protected from nuclease digestion by various DNA-binding peptides and proteins, resulting in a prolonged presence in the circulation and increased pathogenetic activity [42,61]. It has also been reported that a group of human monoclonal antibodies cross-reactive with dsDNA, *Crithidia luciliae*, histone, and apoptotic Jurkat cells protected NETs from digestion via micrococcal nuclease or DNase I. Interestingly, another group of antibodies which were specific to dsDNA did not show significant protection [65]. In a different context, monoclonal antibodies to DNase1L3 protected chromatin from degradation with this enzyme [10]. DNase1L3 is a member of the DNase1 family, which is responsible for the DNase activity in plasma together with DNase1 itself. To make matters still more complicated, these anti-DNase1L3 antibodies are cross-reactive with dsDNA, as described in Section 2 (the generation of anti-DNA antibodies).

#### 4.5.3. Thrombogenic Properties

The prognosis of SLE has improved, and patients now enjoy nearly as long a life expectancy as the average in developed countries. Accordingly, it has become a problem that they suffer a higher cardiovascular disease risk than the general population. One of the causes of this may be the use of corticosteroids over extended periods. In addition, as discussed in Section 4.2.2. (APS) above, NETs lead to a prothrombotic and procoagulant state due to multiple mechanisms. Even in patients with SLE that do not fulfill the criteria for APS, NET-related thrombogenic properties might be relevant to their prognosis. Also, anti-DNA antibodies may contribute directly to cardiovascular risk. Recent in vitro studies demonstrate that anti-dsDNA antibodies purified from SLE patients bind to the cell surface, and some of them enter the nucleus of healthy monocytes, leading to the expression of proinflammatory and prothrombotic molecules, including tissue factors [66]. These results suggest that anti-DNA antibodies, as well as NETs, play a role not only in the pathogenesis of SLE itself but also in associated cardiovascular disorders.

#### 4.5.4. Induction of NET Release by Anti-DNA Antibodies

Several studies have examined whether anti-DNA antibodies induce the release of NETs from neutrophils. For example, mouse monoclonal anti-LL37 and anti-HNP antibodies could induce healthy human neutrophils to release NETs. F(ab^’^)_2_ fragments of anti-LL37 and anti-HNP also induce NET release, suggesting that these antibodies bind neutrophils not via Fc receptors but via cell surface antimicrobial peptides. However, a mouse monoclonal anti-DNA antibody H241 could not induce NET release in this study [42]. In another study, SLE plasma induced NET release using healthy neutrophils [44]. However, patient plasma contains many different antibodies, including immune complexes, cytokines, and other factors. It was not determined which of those was responsible for the release of NET. In a comparative study of plasma NET release activity in SLE and APS, as described in Section 4.2.2. (APS), increased levels of anti-dsDNA antibodies were associated with increased NET release, suggesting that anti-DNA antibodies may be responsible for NET release, but this was not determined either [50]. In a study on pediatric SLE, as described in Section 4.2.1. (SLE), anti-RNP antibodies induced patient neutrophils, but not healthy neutrophils, to die by releasing NETs [43]. Unfortunately, the effect of anti-DNA antibodies was not tested in that study.

In another study described in Section 4.2.3. (AAV), soluble IgG isolated from SLE patient serum did not induce NETosis compared to immobilized SLE-IgG [52]. Recent observations by Patiño-Trives et al. [66] reveal that the incubation of normal neutrophils for 6 h long with the affinity of IgG anti-DNA antibodies purified from SLE sera-induced NETosis. Thus, although evidence has been limited so far, the induction of NETosis by anti-DNA antibodies is likely to be observed when assay conditions are appropriate.

## 5. Conclusions

Some, but not all, anti-DNA antibodies can enter live cells. Apart from immortalized cell lines that tend to show increased endocytosis activity, there have also been reports of the internalization of anti-DNA antibodies by different normal cell types, including monocytes, vascular endothelial cells, and astrocytes. These mechanisms are multifarious, with some antibodies entering via Fc receptor-mediated endocytosis, but other mechanisms are also probable. Some of the antibody enters the nucleus, for which mechanisms remain to be elucidated. In any case, such antibodies are thought to carry nucleotides that can stimulate the cells via TLR-9 or other nucleic acid sensors, resulting in cytokine production or sometimes apoptosis and affecting the pathological condition of SLE.

In the circulation of patients with SLE, neutrophils are primed with IFN-α, and other stimuli and are prone to release NETs following additional triggers, including DNA–anti-DNA immune complexes. NETs are protected from DNase digestion by different proteins, peptides, anti-DNA and other antibodies enveloping the DNA and, therefore, persist for a long period. Such complexes of DNA and proteins/peptides are engulfed by pDCs and macrophages, resulting in the expression of type I IFN, which plays a pivotal role in forming the IFN signature. Thus, a vicious circle is initiated. These accumulative findings indicate the need to formulate a new therapeutic approach targeting anti-DNA antibody production or NET release for the treatment of SLE.

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
