# Peer review of "An Emerging Role for Anti-DNA Antibodies in Systemic Lupus Erythematosus"

_ijms, 2023, doi:10.3390/ijms242216499_

Round 1

Reviewer 1 Report

Comments and Suggestions for Authors

The review provides an overview of the generation and pathogenic role of anti-DNA antibodies in systemic lupus erythematosus (SLE), including the mechanisms by which anti-DNA antibodies penetrate live cells and neutrophil extracellular traps (NETs), and the involvement of anti-DNA antibodies in these processes.

While the text is well-supported by literature from the author group, it could benefit from more specific details on the mechanisms of internalization and the effects of anti-DNA antibodies on each cell type, including monocytes, pDCs, endothelial cells, astrocytes, and neutrophils. Additionally, the text should mention the crucial role of B cells in the generation of anti-DNA antibodies, including the fact that they can be generated from non-autoreactive B cells during a normal immune response and can bind to unique DNA structures and chromatin components.

Line 70-75 on page 2 should be moved to the NETs section, and an abbreviation for HMGB1 is needed on line 166. Furthermore, the text should describe how the complex exits brain endothelial cells on line 154-155 of page 4, and whether this is a common phenomenon in other cells.

Comments on the Quality of English Language

None

Author Response

Reply to Reviewer 1,

Thank you for your valuable comments. I modified my manuscript following your suggestion and showed the changes in red ink in the text.

  • There are many reports in which researchers studied internalization of anti-DNA antibodies by immortalized cell lines that tend to show increased activity of endocytosis. To the contrary, our lab and some other researchers used normal mature cells. Although there have been only limited information about the mechanism and effect of the internalization by normal cells, I added some information to the description of our WB-6 antibody (Line 125 - 129).

  • How DNA-reactive B cells are generated, stimulated and produce anti-DNA antibodies in SLE is a big unanswered question. I added some description of this issue in the text with a relevant review article written by Dr Rekvig in 2015 (Line 49 - 53).

  • Following your suggestion, I moved the description about oxidized DNA as a possible primary antigen for anti-DNA antibody production, to the section of NETs (Line 312 - 314).

  • I added full spelling for HMGB1 (Line 176).

  • How DX1 enters and exits brain endothelial cells has not been determined in their paper, but I found an article which discusses antibody transcytosis across the BBB. I cited the paper and added some description to the manuscript (Line 163, Reference No. 34).

Reviewer 2 Report

Comments and Suggestions for Authors

The review analyzes the role of anti-DNA antibodies in the pathogenesis of Systemic lupus erythematosus (SLE), specifically the mechanisms through which these antibodies penetrate into live cells, and the effect on NETosis. NETS are involved in a variety of other autoimmune diseases, such as rheumatoid arthritis, antiphospholipid syndrome and anti-neutrophil cytoplasmic antibody-associated vasculitis. As a result of entering normal cell types, anti-DNA antibodies influence these cells into releasing specific cytokines and prothrombotic molecules. This subject is relevant for future clinical studies, as the internalization of anti-DNA antibodies may have an effect on pathogenic mechanisms of SLE, with further clinical implications. Thus, NET release or anti-DNA antibody production could represent an important treatment target in the therapeutic strategy of SLE. Taking into account that SLE is a debilitating disease, with multiple complications and important organ damage, this new perspective on treatment is of paramount importance and should be further studied.

The manuscript is well organized, with a logical and succinct structure. It presents a high level of written clarity and a qualitative scientific content that is optimally referenced and rigorously researched. Given the complex nature of the subject, the fluency of the scientific language and the thorough explanation of each relevant concept and mechanism are essential for a high level of comprehensibility. Furthermore, the use of elaborate figures that concisely present the main concepts of the review greatly improves the readability of the manuscript.

After analyzing this manuscript, it can be considered for publication.

Author Response

To Reviewer 2

Thank you very much for your positive estimation. I was encouraged.